# The Roles of Different Fractions in Freshwater Biofilms in the Photodegradation of Methyl Orange and Bisphenol A in Aqueous Solutions

**DOI:** 10.3390/ijerph192012995

**Published:** 2022-10-11

**Authors:** Haojie Yin, Lingling Wang, Guangshu Zeng, Longfei Wang, Yi Li

**Affiliations:** 1Institute of Microbiology, Guangdong Academy of Sciences, State Key Laboratory of Applied Microbiology Southern China, Guangzhou 510070, China; 2Key Laboratory of Integrated Regulation and Resource Development on Shallow Lakes, Ministry of Education, College of Environment, Hohai University, Nanjing 210098, China

**Keywords:** methyl orange, bisphenol A, photodegradation, freshwater biofilm, extracellular polymeric substances, reactive oxygen species

## Abstract

Freshwater biofilms play an important role in the migration and transformation of organic pollutants, especially under illumination conditions. Nonetheless, the roles of variable fractions in freshwater biofilms, e.g., extracellular polymeric substances (EPS), microbial cells and original biofilms, in promoting the photodegradation of trace organic pollutants remain largely unclear. In this study, two contaminants, i.e., methyl orange (MO) and bisphenol A (BPA), were selected, and the roles of different fractions in freshwater biofilms in their photodegradation performances were investigated. After dosing 696 mg/L SS biofilm harvested from an effluent-receiving river, the direct photodegradation rate of MO and BPA was increased 8.7 times and 5.6 times, respectively. River biofilm EPS contained more aromatic fractions, chromogenic groups and conjugated structures than biofilm harvested from a less eutrophic pond, which might be responsible for the enhanced photodegradation process. The quenching experiments suggested that when EPS fractions derived from river biofilm were dosed, ^3^EPS* was the major reactive oxygen species during the photodegradation of MO and BPA. Meanwhile, for EPS derived from the pond biofilm, ·OH/^1^O_2_ was predominantly responsible for the enhanced photodegradation. Batch experimental results suggested that the cells and EPS in river biofilms could collaboratively interact with each other to enhance the preservation of reactive species and protection of microbes, thus facilitating the photoactivity of biofilms. Our results might suggest that biofilms generated from eutrophic waterbodies, such as effluent-receiving rivers, could play a more important role in the photodegradation processes of contaminants.

## 1. Introduction

Bisphenol A (BPA) serves as a representative endocrine disrupting compound, which has been widely used in plastic bottles and food packaging [1,2]. The presence of BPA, even at trace concentrations, could affect the health of the human reproductive system and other tissues. Synthetic dyes have been extensively consumed in papermaking, rubber, leather, textile and other fields, among which azo dyes, such as methyl orange (MO), are the most widely used owing to their bright color and low price [3]. Since traditional sewage treatment facilities could not effectively remove trace organic compounds, such as BPA and MO, the excess presence of these pollutants inevitably entered the natural waterbodies. The concentrations of BPA and MO could reach as high as 16.2 μg/L and 5.0 μg/L [2,4], respectively, in freshwaters, which have been a subject of research concern in recent decades.

Freshwater biofilm, or periphyton, is a complex miniature ecosystem comprising algae, bacteria, protozoa, metazoan and their secrets. Owing to the retention of diverse enzymes and high metabolic activities, freshwater biofilm has been proven effective in accelerating the degradation of a wide range of pollutants [5,6]. By integrating periphyton-based technologies and wastewater treatment theories, researchers have developed a series of technologies, including biofiltration, artificial biological gaskets and microbial immobilization technology, which have found applications in controlling non-point source pollutions and algal blooms [7,8]. The removal mechanisms of trace organic pollutants by freshwater biofilms include adsorption, bioaccumulation, biodegradation and photodegradation under illumination conditions. Freshwater biofilms exhibit high adsorption and accumulation capacities with a wide range of pollutants, including heavy metals, active pharmaceutical ingredient, endocrine disrupting compounds, etc. [6]. Notably, recent studies have confirmed that freshwater biofilms could play an important role in the migration and transformation of organic pollutants, especially under illumination conditions [8,9].

The presence of a certain amount of reactive species, including hydrogen peroxide (H_2_O_2_), hydroxyl radicals (·OH) and singlet oxygen (^1^O_2_), has been reported in previous studies. During the electron transfer processes in algal cells, the production of reactive species, such as ^1^O_2_, could induce the production of H_2_O_2_ via disproportionate reactions [10]. The detected concentration of H_2_O_2_ in a freshwater biofilm system could reach as high as 40.1 μmol/L [11,12]. The produced reactive species have been proven a key factor in accelerating the degradation of various organic pollutants, including trichloroacetaldehyde, atrazine and sodium dodecyl benzene sulfonate, in a freshwater biofilm system [13,14,15]. Despite freshwater biofilms being verified as efficient in promoting the degradation of pollutants, the roles of variable reactive species in the degradation process, as well as the factors governing the production of reactive species, still remain to be explored.

Extracellular polymeric substances (EPS), comprising proteins, polysaccharides, humic substances, DNA debris and other microbial secretions, serve as an important protective barrier to freshwater biofilms [16,17]. Studies have confirmed that algae and their secretions played an important role in organic contaminant photodegradation through screening the sunlight or initiating photolysis [18,19]. The EPS fractions derived from algae could enhance the solar photodegradation of chlortetracycline and remarkably alter the degradation pathways via photolysis [18,20]. The photoproduction of reactive species, including triplet excited-state species, hydroxyl radical, singlet oxygen and superoxide, have been proven as being involved in the enhanced degradation of organic contaminants [21,22]. The triplet excited-state species, i.e., ^3^EOM^*^, can not only react with contaminants directly but can also be a precursor of reactive species, such as ^1^O_2_. Studies have found that extracellular organic materials originating from different algae sources varied in chemical properties, which might result in different yields of ^3^EOM^*^ [20,21]. Nonetheless, the information regarding the photosensitivity of different extracellular organic materials originating from different sources of freshwater biofilms still remains unclear.

Despite the fact that previous studies have explored the removal mechanism of trace organic contaminants induced by freshwater biofilms under illumination, the roles of variable fractions in freshwater biofilms, i.e., EPS, microbial cells, as well as raw biofilms, in promoting the photodegradation of organic contaminants remain largely unclear. The roles of EPS and microbial cells during the removal of contaminants need to be clarified. The properties of freshwater biofilms, the compositional characters and the capacity to remove trace organic contaminants have been proven remarkably associated with the origins of the biofilms [23,24,25]. An investigation on how freshwater biofilms from distinct cultivation sources behaved differently in enhancing the photodegradation of trace organic contaminants will be beneficial to understanding the decaying mechanisms of trace organic contaminants in real scenarios. Effluent-receiving rivers represent one of the most typical ecosystems heavily impacted by anthropogenic activities, which contain a certain amount of trace organic contaminants with different physicochemical properties [26]. Whether the freshwater biofilms cultivated in effluent-receiving rivers behave differently in the photodegradation of trace organic contaminants needs to be clarified.

In the present study, two representative trace organic contaminants with high detection rates and relatively high concentrations in natural aquatic systems, i.e., MO and BPA, were selected for investigation. First, freshwater biofilms from two freshwaters with distinct water properties were selected, and their effects on the photodegradation behaviors of MO and BPA were evaluated. Secondly, the roles of different fractions in freshwater biofilms, i.e., raw biofilm, biofilm with EPS and biofilm without EPS, in the photodegradation of contaminants were investigated. The roles of variable reactive species in the degradation processes were explored. Lastly, the photosensitizing mechanism of freshwater biofilms regarding the variable biofilm fractions in pollutant degradation was elucidated. The study is of significance in understanding the ecological functions of freshwater biofilms, as well as in providing data support for predicting the retention time and ecological risk of trace organic pollutants in aquatic systems.

## 2. Materials and Methods

### 2.1. Sampling of Freshwater Biofilms

Freshwater biofilms were collected from two representative freshwaters with distinct water qualities. The first sampling site is located at an effluent-receiving river of the Lintong District WWTP in Xi’an, China (34.401° N, 109.205° E) and uses a sequencing batch reactor (SBR) process that treats a volume of 25,000 m^3^ wastewater from domestic sources. The discharged effluent occupies almost all the water flow in the receiving river. The second sampling site is located in a natural landscape pond in Nanjing, China (32.062° E, 118.767° N) and receives natural precipitation as its main water source.

Freshwater biofilms on the river banks or cobblestones were carefully scrubbed with a sterile brush and collected. The bulk biofilm suspensions were filtered through a 0.85 mm pore size screen to remove the larger particles, and the solid substance concentrations of the bulk biofilm solutions were measured prior to extracting the EPS [27,28]. The two sampled freshwater biofilms were named ‘River Biofilm’ and ‘Pond Biofilm’, respectively, in the following manuscript.

The physical and chemical parameters of the water samples from both sites, including pH, conductivity and redox potential, were measured in situ using a Water Quality Checker U-10 (HORIBA). Ammonium nitrogen (NH_3_-N), total nitrogen (TN), total phosphorus (TP) and chemical oxygen demand (COD) were measured to reflect the trophic conditions at the sampling sites. Three replicates were used for all parameters tested (Appendix A). The values of COD, NH_3_-N, TN and TP for the River Biofilm sampling site were 23.45, 1.18, 9.66 and 0.12 mg/L, respectively, while the values were 17.62, 1.03, 1.57 and 0.10 mg/L at the Pond Biofilm sampling site. Appendix A clearly suggests a more eutrophic status of the water environment in the effluent-receiving river with higher organic carbon concentration and nitrogen level.

### 2.2. Chemicals, EPS Extraction and Characterization

The standard chemicals of orange methyl (OM) and bisphenol A (BPA) (>98%) were purchased from Sinopharm Chemical Reagent Co., Ltd. (Shanghai, China). NaN_3_, isopropanol and sorbic acid were purchased from Sigma-Aldrich. The other chemicals used in the study were purchased from Sinopharm Chemical Reagent Co., Ltd.

The suspended solid contents of two bulk biofilm suspensions were measured following the standard protocol [29], and both suspensions were diluted to a mixture containing 696 mg/L SS for the following experiments. A modified heat extraction protocol was employed to extract the EPS fractions from both freshwater biofilms. Briefly, the mixtures containing 696 mg SS/L freshwater biofilms were centrifuged at 4500× *g* at 4 °C for 5 min three times to remove residual liquids. Then, the samples were re-suspended in 50 mM phosphate-buffered solution and heated to 60 °C in a water bath for 30 min. Afterward, the solution was centrifuged at 10,000 rpm for 15 min again. After filtration through a 0.45 μm membrane, the supernatant was stored at 4 °C, and it was used as the EPS solution for the following experiments.

### 2.3. Characterization of EPS Fractions

The DOC concentrations of the EPS solutions in this study were measured using a multi N/C2100 TOC analyzer. The concentrations of the proteins and polysaccharides in the EPS solutions (both with a TOC content of 50 mgC/L) were measured using a modified Lowry method and the anthrone method, with egg albumin and glucose as the standard solutions, respectively [30]. UV-vis spectra of EPS solutions at TOC concentrations of 10 mgC/L and 20 mgC/L were recorded using a UV-visible spectrophotometer (DR6000, HACH, Loveland, CO, USA). UV_254_ absorbance per mg C was calculated as the SUVA values, and the ratios of UV_250_ and UV_365_ were calculated as E2/E3 ratios. Three-dimensional EEM fluorescence spectra were obtained using a luminescence spectrometer (F-7000 FL, Hitachi, Tokyo, Japan). The spectra were recorded with scanning emission spectra from 250 to 500 nm at 5 nm increments by varying the excitation wavelength from 200 to 450 nm at 5 nm increments. The Milli-Q water spectrum was collected as the background. We also calculated parameters including the biological index (BIX), humification index (HIX) and fluorescence index (FI) to describe the fluorescent characteristics of biofilm EPS, as described in Appendix A.

The specific chemical structures of the EPS were analyzed by the nuclear magnetic resonance spectrometer (NMR) (Agilent Vnmrs 600 NMR Spectrometer). The relative contents of the chemical functional groups were analyzed via an X-ray photoelectron spectrometer (XPS) (Thermo Scientific ESCALAB 250Xi 1600 spectrometer). Based on the significant peaks of functional groups (e.g., 284.8 eV for carbon chain and hydrocarbyl (i.e., C-(C/H)), 286.43 eV for epoxy and alkoxy (i.e., C-(O/ N)) and 288.66 eV for ester groups (i.e., O-C=O))), the XPS C1s spectra data were statistically fitted and analyzed by the Origin 2017 software. Then, the oxygen-containing functional groups’ composition was estimated using the area of O-C/O-H and C=O. All samples were prepared using a freeze-drying pretreatment [18,20].

### 2.4. Photochemical Reaction Experiments

Prior to the photochemical reaction experiments, stock solutions containing 100 mg/L BPA or ethyl orange were prepared using distilled water. The irradiation experiments were performed using a device equipped with a 300 W Xenon lamp (HDL-II, Bobei, China) emitting a wavelength between 290 and 600 nm to mimic natural sunlight by using an optical filter to remove light at a wavelength of 280 nm. The average irradiation density was 30.2 mW/cm^2^, and the emitting wavelength of the irradiation system was slightly richer in UV components than the sunlight (295–400 nm). The light source was set 20 cm beneath the liquid surface, and the photolysis experiments were maintained at 25 ± 0.5 °C in a water bath (Appendix A). The photochemical reactor was set as a 100 mL beaker with a magnetic stirring apparatus. The magnetic paddle was kept stirring at a speed of 60 rpm to maintain a uniform mixture of reactants.

Firstly, the effects of biofilm addition on the photochemical reactions of the pollutants were evaluated. To amplify the effects of photochemical reactions, a concentration higher than the values reported in synthesized water samples was applied in this study, following the protocol of several studies [18,31]. Briefly, 200 mL of 10 mM OM or BPA solution (10 mM PBS, pH 6.8) was held in the aforementioned beaker reactors. Stock biofilm solutions were added to the beaker to reach a SS concentration of 696 mg/L (with a corresponding EPS content of 14 mgC/L, following the protocol in Section 2.2). Thereafter, variable amounts of bulk pollutant stock solutions (100 mg/L) were added to the beaker to obtain solutions containing pollutant concentrations of 1.0, 2.0 and 5.0 mg/L, respectively. To evaluate the effects of initial concentrations of freshwater biofilms on the photochemical reactions, the experimental groups containing 2 mg/L of MO and BPA were selected. The initial concentrations of freshwater biofilms were set as 348 mg/L and 696 mg/L, respectively.

To evaluate the effects of different biofilm fractions, i.e., raw biofilm, biofilm with EPS and biofilm without EPS, on the photochemical reactions of the pollutants, as shown in Figure 1, 100 mL of bulk biofilm suspension with a SS content of 696 mg/L was added to reactor 1. Another 100 mL of bulk biofilm suspension was utilized for EPS extraction, following the protocol in Section 2.2. After EPS extraction, EPS solutions and biofilm without EPS were re-suspended in distilled water in a volume of 100 mL. The DOC analysis of the EPS solutions suggested that 696 mg/L EPS could acquire an EPS solution with a concentration of 14 mg C/L. Afterward, several drops of MO/BPA stock solutions were added to each beaker to reach a content of 2 mg/L. All the degradation experiments were performed three times.

### 2.5. Reactive Species Detection

To investigate the role of reactive species (i.e., ·OH, ^1^O_2_ and ^3^EOM^*^ et al.) in the process of indirect pollutant photodegradation induced by EPS fractions, reactive species were selectively quenched by a scavenger (e.g., NaN_3_ for ·OH and ^1^O_2_, isopropanol for ·OH, and sorbic acid for ^3^EOM^*^, respectively) [18,20,32]. In each batch test, isopropanol, NaN_3_ and sorbic acid were added to the target mixture with a concentration of 2.0 mM, 100.0 mM and 2.0 mM, respectively, prior to photochemical reaction experiments [31]. All experiments were performed three times.

### 2.6. Analytical Methods

For each photochemical reaction experiment, aliquots of 2.0 mL filtrated samples were collected at different time intervals, i.e., 0, 5, 10, 15, 30, 45 and 60 min, from the solution. The content of MO was measured via UV spectra at a wavenumber of 465 nm. The contents of BPA were determined by high-performance liquid chromatography (1260 Infinity, Agilent Co., Santa Clara, CA, USA) equipped with a 5 μm × 4 mm × 150 mm Eclipse XDB-C18 column set at 35 °C and a UV detector at a wavenumber of 230 nm. The mobile phase was a mixture of 10 mM H_2_O/acetonitrile 50:50 (*v/v*) at a flow rate of 1 mL/min. The photodegradation rate constants of MO and BPA followed by pseudo-first-order depletion were obtained by linear regression, following Equation (1):(1)$$\ln\frac{C_{0}}{C}=K_{\mathrm{obs}}t$$
where *K*_obs_, *t*, *C* and *C*_0_ refer to the pseudo-first-order depletion constant (min^−1^), reaction time (min), pollutant concentration during reaction (mg/L) and initial concentration of the pollutant (mg/L), respectively.

## 3. Results and Discussion

### 3.1. Photodegradation of Pollutants in the Presence of Freshwater Biofilms

Figure 2 shows the photodegradation behaviors of MO and BPA at 2 mg/L in aqueous solutions in the absence of freshwater biofilms. Limited declines in the concentrations of pollutants were observed under both dark and illumination conditions, suggesting that MO was not easily photodegraded under simulated natural lighting conditions in Figure 2a. After dosing river biofilm suspensions at a dry biomass concentration of 696 ± 45 mg/L into the beakers, the removal efficiency of MO was increased by 36.0% after 30 min of illumination. The direct photodegradation rate of OM increased remarkably from 0.0018 min^−1^ in the dark conditions to 0.0157 min^−1^ under illumination. Similar to the results of MO, limited declines in BPA concentrations were observed in the presence of river biofilms under dark conditions, as shown in Figure 2b, implying that physical and chemical processes, i.e., hydrolysis and adsorption, were insignificant. After dosing the same amount of river biofilm suspensions, the removal efficiency of BPA was increased by 19.8% after 2 h. The direct photodegradation rate of BPA was increased from 0.0012 min^−1^ in the absence of biofilms to 0.0067 min^−1^ after biofilm addition.

The effects of the initial pollutant concentration on the photodegradation behaviors were evaluated, as shown in Appendix A. Appendix A shows that after adding 696 ± 45 mg/L river biofilm suspensions, the direct photodegradation rate of OM was decreased from 0.0181 min^−1^ at the initial concentration of 1 mg/L to 0.0037 min^−1^ when the initial concentration was increased to 5 mg/L. In comparison, a smaller impact of BPA photodegradation rate was detected when the initial concentration was increased from 1 mg/L to 5 mg/L, with a k_obs_ value decreasing from 0.0021 min^−1^ to 0.0018 min^−1^.

Figure 3 shows the impacts of freshwater biofilm sources and concentrations on the photodegradation of OM. After dosing 348 mg/L river biofilm suspensions, a BPA degradation efficiency of 27.5% was observed at 30 min of illumination, almost five times the value in the absence of biofilms. Meanwhile, in the system dosed with 696 mg/L river biofilm suspension, the degradation efficiency was increased to 41.5% in 30 min. In comparison, the OM degradation efficiency in the presence of the pond biofilm was much lower than when dosed with the river biofilm, as shown in Figure 3b, with the values of 6.7% and 14.5%, respectively, when dosed with concentrations of 348 and 696 mg/L. Similar results were also observed when treating BPA-containing aqueous solutions.

Appendix A summarizes the direct photodegradation rates of pollutants at a concentration of 2 mg/L. In aqueous solutions dosed with river biofilms, the *k*_obs_ values were much higher than in the absence of biofilms, increasing from 0.17 × 10^−2^ min^−1^ to 1.18 × 10^−2^ min^−1^ and 1.55 × 10^−2^ min^−1^, respectively, at a dosing content of 348 mg/L and 696 mg/L. Meanwhile, in aqueous solutions dosed with pond biofilms, the amplification in degradation efficiency after biofilm dosing was much smaller than with river biofilm dosing, with *k*_obs_ values of 0.23 × 10^−2^ min^−1^ and 0.46 × 10^−2^ min^−1^, respectively, at a dosing content of 348 and 496 mg/L. Similar results were also observed in the case of BPA degradation behaviors, as shown in Appendix A.

### 3.2. Roles of ROS during Photodegradation of Pollutants in the Presence of Biofilms

To explore the roles of different reactive oxygen species, i.e., ·OH, ^1^O_2_ and ^3^EOM^*^, in the indirect photodegradation of target pollutants in the presence of freshwater biofilms, ROS were, respectively, quenched by the corresponding scavengers, i.e., NaN_3_ for ·OH and ^1^O_2_, isopropanol for ·OH and sorbic acid for ^3^EOMs*, respectively [33,34]. As shown in Figure 4a, in the presence of isopropanol or sorbic acid, a suppression of MO photodegradation was detected, demonstrating that the reaction of ^1^O_2_ and ^3^EOMs* with MO made a contribution to the total degradation. Notably, a remarkable suppression of photodegradation was observed in the presence of NaN_3_, suggesting that the enhancement influences of river biofilms on MO degradation were mainly ascribed to the reaction between MO and ·OH/^1^O_2_.

Figure 4b shows the photodegradation rate of MO with quenchers in the presence of the pond biofilm. The presence of isopropanol or NaN_3_ hardly impacted the photodegradation behaviors of MO and even suppressed the degradation efficiencies in the early stages. The presence of sorbic acid significantly decreased the MO photodegradation rate, suggesting the reaction between MO and ^3^EOMs* was mainly responsible for the total photodegradation. Appendix A illustrates the photodegradation rate of BPA with quenchers in the presence of freshwater biofilms. Similar suppression behaviors in photodegradation efficiencies were observed, suggesting that ·OH /^1^O_2_ and ^3^EOMs* served as the major reactive oxygen species in the photodegradation processes in the presence of river biofilm and pond biofilm, respectively.

### 3.3. Characterization of EPS Derived from Freshwater Biofilms

Appendix A provides the compositional properties of EPS derived from both freshwater biofilms. For an EPS solution at a TOC concentration of 50 mgC/L, the EPS derived from the river biofilm contained 94.3 ± 16.4 mg/L carbohydrates, 13.2 ± 0.2 mg/L proteins and 26.7 ± 0.1 mg/L humic substances, respectively. Meanwhile, for the pond biofilm EPS at the same TOC content, the concentrations of carbohydrates, proteins and humic substances were 72.6 ± 27.5 mg/L, 4.8 ± 1.0 mg/L and 15.5 ± 0.7 mg/L, respectively. The contents of proteins and humic substances in river biofilm EPS were 2.75 and 1.73 times higher than in pond biofilm EPS.

An integrated spectral approach was employed to decipher the differences between EPS fractions derived from both freshwater biofilms. Appendix A provides the spectral parameters of both EPS fractions. River biofilm EPS had higher a(355), SUVA_254_ and SUVA_260_ values than pond biofilm EPS, implying that the EPS from the biofilm collected in effluent-dominated rivers contained more fractions with high aromaticity, and the fractions were more hydrophobic [35,36]. The value of E_2_/E_3_ was reported to be significantly negatively correlated with the apparent molecular weight and aromatic condensation degrees. The higher E_2_/E_3_ value of pond biofilm EPS suggested that the EPS molecules derived from natural pond biofilm contained more high-molecular-weight fractions and aromatic condensation degrees. River biofilm EPS had a higher HIX value compared to those of the pond biofilm EPS. The results suggested that river biofilm produced more humified materials [37]. The BIX value of river biofilm EPS was in the range of 0.8 to 1, demonstrating that the biopolymers originated predominantly from an autochthonous source. Meanwhile, the BIX of pond biofilm EPS was higher than 1, suggesting that the structural plant, bacterial and algal residues comprised the major sources.

Figure 5 shows the fluorescence spectra of EPS derived from both biofilms. Two peaks of both EPS, located at Ex/Em 220/350 nm (Peak A) and 275/345 nm (Peak B), were, respectively, associated with aromatic-protein-like and tryptophan-protein-like fractions [38]. The major differences in the spectra centered at Peak B, suggesting the EPS derived from biofilms collected in effluent-dominated rivers contained a higher amount of tryptophan-protein-like fractions.

The freeze-dried EPS samples were analyzed by 13C NMR to characterize the chemical functional groups. Table 1 demonstrates a relatively large contribution of alkyl (0–45 ppm) and carbohydrate (63–93 ppm) fractions, ranging from 34.6% to 41.3% and from 37.2% to 38.1%, respectively. Table 2 shows the contributions of each functional group in both EPS fractions via NMR analysis, including the alkyl, methoxyl, carbohydrate, aryl, O-aryl, carboxyl, carbonyl, aromatic carbon, aliphatic carbon and polar carbon fractions. Compared with pond biofilm EPS, the EPS derived from river biofilm contained higher amounts of aromatic carbon (7.9% compared to 2.1%) and polar carbon (61.6% compared to 56.5%), suggesting that river biofilm EPS were more aromatic and hydrophilic, consistent with the results of spectral analysis in Figure 5 and Appendix A. The peak fitted results of C1s spectra in Figure 5 implied that the EPS from both freshwater biofilms contained two similar peaks and a shoulder peak, i.e., 284.8 eV for C-(C/H), 286.43 eV for C-(O/N) and 288.66 eV for O-C=O, respectively. After statistical fitting, it was found that the contents of carbon chain/hydrocarbyl contributed to 78.49% of the carbon-containing structure of River Biofilm EPS, higher than those in Pond River EPS, as shown in Table 2. The higher amount of hydrocarbyl compounds in EPS fractions may result in higher photo-reactivity. In previous studies, the carbonyls and ketone/quinone compounds have been proven to play an important role in photoreactivity of DOM fractions, such as humic substances [39]. In comparison, the epoxy/alkoxy/amino groups and ester groups contributed to 12.27% and 9.24% of the carbon-containing structure of River Biofilm EPS, lower than those in Pond River EPS.

### 3.4. Roles of Different Fractions in Biofilms in the Photodegradation of Pollutants

To reveal the different roles of biofilm fraction, i.e., raw biofilm, biofilm without EPS, as well as biofilm with EPS, in the photodegradation behaviors of target pollutants, the raw biofilms were pre-treated, following the protocols detailed in Section 2.4. The photodegradation behaviors of variable fractions of freshwater biofilms are shown in Figure 6 and Appendix A.

Figure 6 shows the different roles of ROS during the photodegradation of MO. In the presence of river biofilm EPS, as shown in Figure 6a, a photodegradation efficiency of 10.3% was observed after 30 min of illumination. After dosing of sorbic acid, the overall photodegradation efficiency decreased to 1.6%, implying a suppression efficiency of 84%. Meanwhile, in the presence of pond biofilm EPS shown in Figure 6b, the highest photodegradation suppression was also observed by adding sorbic acid, and the dose of NaN_3_ and IPA hardly impacted the overall photodegradation efficiencies. Similar results were also observed in batch experiments during the degradation of BPA, as shown in Appendix A.

To explore the different roles of photosensitized ROS in the photodegradation processes, quenching experiments were employed, as shown in Figure 7. After adding different quenchers into the batch experiments containing river biofilm EPS, ^1^O_2_, ∙OH and the triplet excited-state EPS species (^3^EPS^*^) were confirmed as being involved in the photodegradation process of MO. The addition of sorbic acid decreased remarkably the overall photodegradation efficiency, suggesting that the interaction between MO and ^3^EPS^*^ played a major role during the process. In comparison, the dose of NaN_3_ and IPA made a small contribution to the photodegradation suppression, and the difference was insignificant, as shown in Figure 7b. Meanwhile, the addition of sorbic acid even promoted MO degradation, which was probably due to the light shielding effects of DOM molecules or their interactions with pollutants. Previous studies have confirmed that the photoreactivity of DOM and the photodegradation behaviors of target pollutants varied remarkably among DOM originating from different sources [18,20,39]. The distinct photodegradation behaviors of OM in the present study were probably associated with the distinct properties of EPS, which will be discussed in the following sections of the work. Similarly, the photodegradation behaviors of BPA were also evaluated, and ^3^EPS^*^ were proven as playing a vital role in the photodegradation of BPA induced by biofilm EPS, as shown in Appendix A.

Algae serve as the predominant microbial fractions in freshwater biofilms, whose role in the photodegradation of emerging contaminants, e.g., antibiotics, EDCs, etc., has attracted research interest in recent years. Zhang et al. investigated the photodegradation behaviors of norfloxacin in the presence of Chlorella and reported an accelerated degradation efficiency of norfloxacin induced by the generated hydroxyl radical after illumination of UV irradiation [40]. The increase in algal concentration led to a higher photodegradation efficiency of norfloxacin. Tian et al. investigated the photodegradation behaviors of aureomycin in the presence of Chlorella vulgaris and observed a degradation efficiency of nearly 96% under irradiation conditions [20]. The studies have proven the significant roles of DOM or EPS fractions generated from diverse sources (e.g., pure strains and activated sludge) in the photodegradation processes of a wide range of pollutants, including antibiotics, active pharmaceutical ingredients, endocrine disrupting compounds, etc. [18,20,31].

### 3.5. Photosensitizing Mechanism of Different Biofilm Fractions in Pollutant Degradation

Figure 8 compares the different *k*_obs_ values during the degradation of OM and BPA in the presence of different fractions in biofilms. The roles of different fractions in biofilms in the direct photodegradation rates during photodegrading MO are shown in Appendix A. After dosing pond biofilm and river biofilm (696 mg/L), the *k*_obs_ values of OM degradation were increased 2.54 and 8.67 times, respectively, as shown in Figure 8a. In comparison, when EPS derived from both biofilms was added into the solutions, pond biofilm EPS (14 mgC/L) hardly impacted the *k*_obs_ value. Meanwhile, the value was increased 2.08 times when river biofilm EPS (14 mgC/L) was dosed. The results implied that the water qualities in cultivating biofilms, as well as the properties of freshwater biofilms, could remarkably impact the photodegradation behaviors of trace organic contaminants. Similar results were observed in the cases of photodegrading BPA by variable fractions of freshwater biofilms, as shown in Figure 8b.

The results might probably suggest that the freshwater biofilms cultivated in effluent-receiving rivers might behave differently in promoting the photodegradation of trace organic contaminants. Wastewater treatment facilities increasingly release a multitude of micropollutants, nutrients and organic compounds to the receiving environments. Studies have found that excess pollution may have pervasive impacts on biodiversity and ecosystem services [23,26]. Figure 8 demonstrates that the *k*_obs_ value (0.01576 min^−1^) induced by river biofilm was much higher than the sum (0.01008 min^−1^) of the two *k*_obs_ values induced by river biofilm with EPS and river biofilm without EPS. The results might probably suggest that biofilms generated from eutrophic waterbodies, such as effluent-receiving rivers, could play a more important role in the photodegradation processes of contaminants.

In previous studies, the protective roles of EPS in enhancing the ecological functions of freshwater biofilms, such as higher tolerance to antibiotics, heavy metals and nanoparticles, have been confirmed [41,42,43]. The present results suggest that the cells and extracellular polymeric substances in biofilms cultivated in these waterbodies could interact with each other to enhance the functions of biofilms, such as the retention of enzymes, preservation of reactive species and protection of microbes [7,17]. Additionally, the higher autotrophic biomass, e.g., Cyanobacteria, and higher biofilm thickness of freshwater biofilms could limit the penetration of illumination, thus facilitating the electron transfer processes in algal cells by eliminating the oxidative damage [44,45]. As a result, the overall photodegradation capacity was higher in a raw biofilm system rather than the sum of the performances when illuminating EPS and biofilms without EPS, respectively, especially for the contaminant of MO. In future studies, the underlying mechanisms of enhanced photodegradation of trace organic contaminants induced by variable fractions in freshwater biofilms harvested in eutrophic waterbodies need to be clarified.

## 4. Conclusions

The two target freshwater biofilms could accelerate the photodegradation of both pollutants. The direct photodegradation rate of MO and BPA was increased 8.7 times and 5.6 times, respectively, when dosed with river biofilms. River biofilm EPS contained more aromatic fractions, chromogenic groups and conjugated structures, which might be responsible for the enhanced photodegradation process. When EPS fractions derived from biofilm harvested from an effluent-receiving river were dosed into the system, ^3^EPS^*^ were proven as the major reactive oxygen species during the photodegradation of MO and BPA. Meanwhile, for EPS derived from pond biofilm, ·OH /^1^O_2_ was predominantly responsible for the enhanced photodegradation performance. The cells and extracellular polymeric substances in biofilms cultivated in more eutrophic waterbodies could collaboratively interact with each other and thus facilitate the overall photodegradation performance of pollutants. Future studies on elucidating the mechanism of enhanced photodegradation of trace organic contaminants induced by variable fractions in freshwater biofilms need to be clarified.

## Figures and Tables

**Figure 1 ijerph-19-12995-f001:**
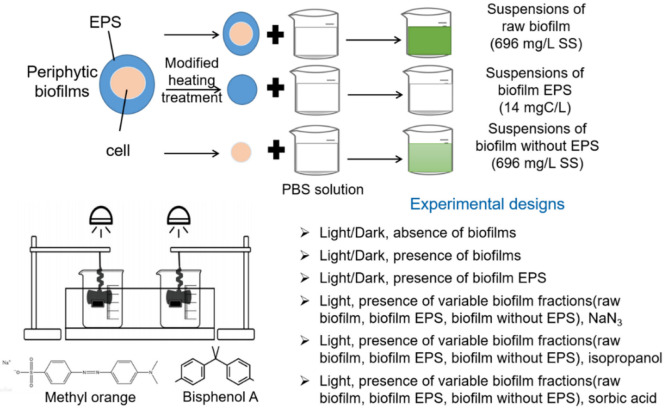
Batch experimental design.

**Figure 2 ijerph-19-12995-f002:**
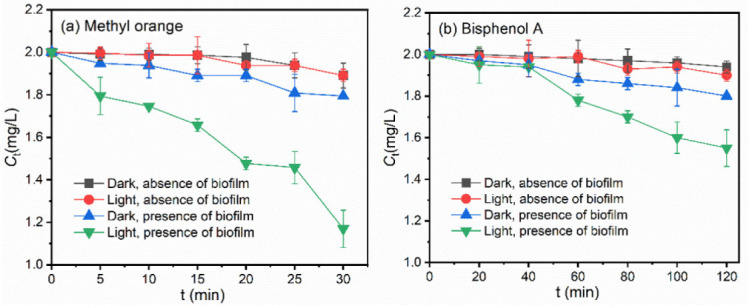
Effects of illumination and biofilm addition on the concentrations of (**a**) MO and (**b**) BPA in aqueous solutions.

**Figure 3 ijerph-19-12995-f003:**
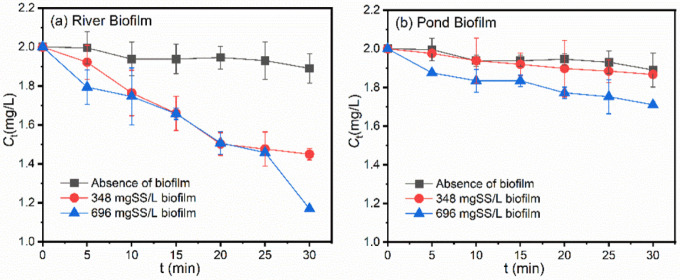
Impacts of (**a**) River biofilm and (**b**) Pond biofilm and concentrations on photodegradation of OM at a concentration of 2 mg/L in aqueous solutions.

**Figure 4 ijerph-19-12995-f004:**
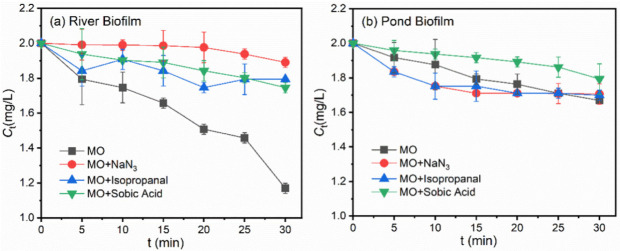
Photodegradation rate of MO with quenchers (isopropanol, NaN_3_ and sorbic acid) in the presence of the suspensions derived from (**a**) River Biofilm and (**b**) Pond Biofilm. Experimental conditions: [MO]_0_ = 2 mg/L, [Biofilm] = 696 mg/L, pH = 6.8 (10 mM phosphate buffer).

**Figure 5 ijerph-19-12995-f005:**
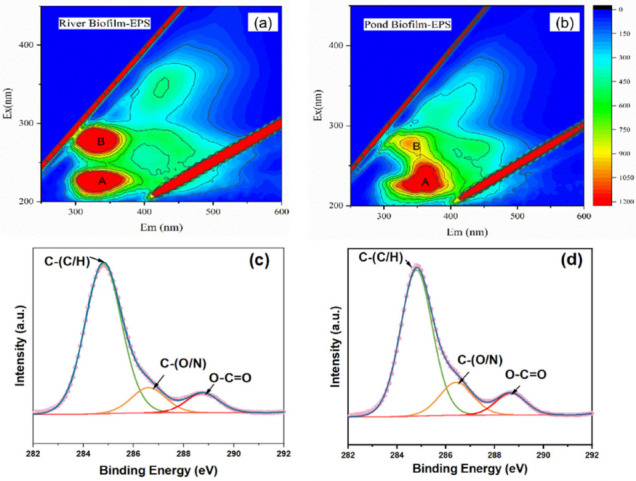
Fluorescence excitation–emission matrices of (**a**) River Biofilm EPS and (**b**) Pond Biofilm EPS. Additionally, XPS spectra for carbon (C1s) of (**c**) River Biofilm EPS and (**d**) Pond Biofilm EPS.

**Figure 6 ijerph-19-12995-f006:**
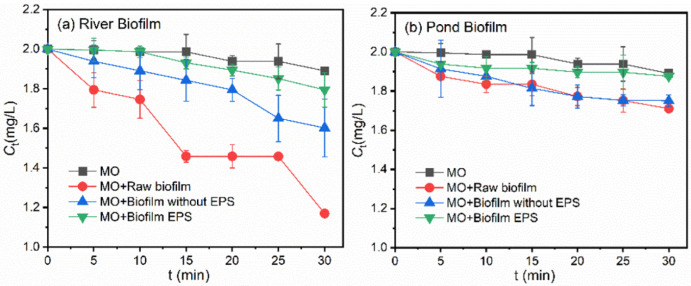
Photodegradation behaviors of MO in the presence of different (**a**) River biofilm fractions and (**b**) Pond biofilm fractions, i.e., raw biofilm with a content of 696 mg/L, the same content of raw biofilm after EPS extraction, as well as the extracted EPS (~14 mgC/L TOC). C_0_[MO] = 2 mg/L.

**Figure 7 ijerph-19-12995-f007:**
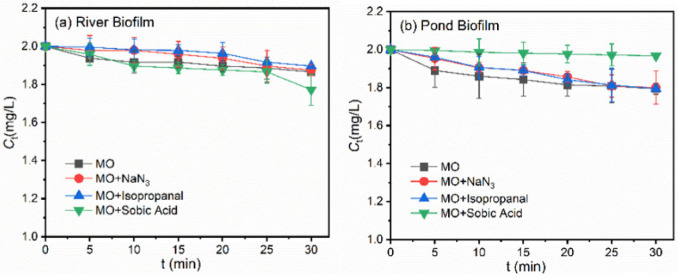
Photodegradation rate of MO with quenchers (isopropanol, NaN3 and sorbic acid) in the presence of the EPS derived from (**a**) River Biofilm and (**b**) Pond Biofilm. Experimental conditions: [MO]_0_ = 2 mg/L, [EPS content] = 14 mgC/L, pH = 6.8 (10 mM phosphate buffer).

**Figure 8 ijerph-19-12995-f008:**
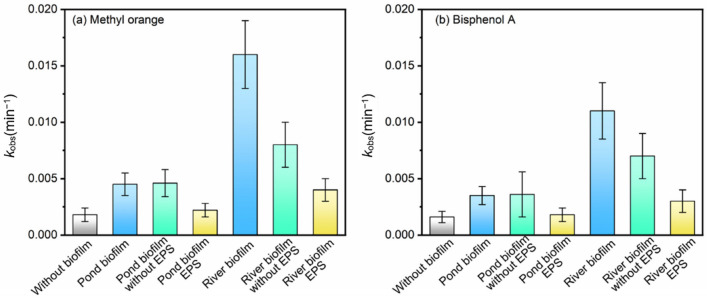
Comparison of kobs values of (**a**) OM and (**b**) BPA in the presence of different fractions in biofilms.

**Table 1 ijerph-19-12995-t001:** Characterization of carbon-containing functional groups of EPS via XPS analysis.

Sample	Carbon-Containing Functional Group Contribution (Area, %)		
	C-(C/H) (i.e., 284.8 eV)	C-(O/N) (i.e., 286.43 eV)	O-C=O (i.e., 288.66 eV)
River biofilm EPS	78.49	12.27	9.24
Pond biofilm EPS	72.71	16.98	10.31

**Table 2 ijerph-19-12995-t002:** Integration results of solid-state ^13^C nuclear magnetic resonance of biofilm samples.

Sample	Alkyl (%) 0–45 ppm	Methoxyl (%) 45–63 ppm	Carbohydrate (%) 63–93 ppm	Aryl (%) 93–148 ppm	O-Aryl (%) 148–165 ppm	Carboxyl (%) 165–190 ppm	Carbonyl (%) 190–220 ppm	Aromatic Carbon ^a^	Aliphatic Carbon ^b^	Polar Carbon ^c^
River biofilm EPS	34.6	12.5	38.1	3.8	4.1	6.6	0.3	7.9	85.2	61.6
Pond biofilm EPS	41.3	14	37.2	2.1	0	5.4	0	2.1	92.5	56.6

^a^ Aromatic Carbon = Aryl (93–148) + O-Aryl (148–165); ^b^ Aliphatic Carbon = Alkyl (0–45) + Methoxyl (45–63) + Carbohydrate (63–93); ^c^ Polar Carbon = Methoxyl (45–63) + Carbohydrate (63–93) + O-Aryl (148–165) + Carboxyl (165–190) + Carbonyl (190–220).

## Data Availability

The data presented in this study are available at this article and Appendix A.

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
