# Peer review of "The Roles of Different Fractions in Freshwater Biofilms in the Photodegradation of Methyl Orange and Bisphenol A in Aqueous Solutions"

_ijerph, 2022, doi:10.3390/ijerph192012995_

Round 1
Reviewer 1 Report
This work explored the removal mechanism of trace organic contaminants induced by two types freshwater biofilms under illumination. The topic is interesting, and the results may provide valuable implications for control of trace organic pollutants in rivers. I would like to recommend its publication after addressing the following minor issues.
1. Introduction: What is the prevailing view on the mechanism of removal of trace organic pollutants by biofilm?More detailed information is suggested to be provided with references to better support the authors' statements.
2. Materials and Methods: The two biofilms come from different regions. Are there congenital differences between the two biofilms due to their geographical location? In the view of the reviewer, the congenital difference may attenuate the effect of water quality?
3. Results and Discussion: Please appropriately add data description and analysis of Table 1 and Table 2 in section 3.3.
4. Please double-check the format and further improve the language.
Author Response
This work explored the removal mechanism of trace organic contaminants induced by two types freshwater biofilms under illumination. The topic is interesting, and the results may provide valuable implications for control of trace organic pollutants in rivers. I would like to recommend its publication after addressing the following minor issues.
- Introduction: What is the prevailing view on the mechanism of removal of trace organic pollutants by biofilm? More detailed information is suggested to be provided with references to better support the authors' statements.
We appreciate the reviewer’s valuable comment. The removal mechanism of trace organic pollutants by freshwater biofilms include adsorption, bioaccumulation, biodegradation, and photodegradation under illumination condition. The related expressions have been added in the manuscript marked in red.
In the revised version, the related expressions have been shown as below:
The removal mechanism of trace organic pollutants by freshwater biofilms include adsorption, bioaccumulation, biodegradation, and photodegradation under illumina-tion condition. Freshwater biofilm exhibit high adsorption and accumulation capacities to a wide range of pollutants including heavy metals, active pharmaceutical ingredient, endocrine disrupting compounds, etc. [6]. Notably, recent studies have confirmed that freshwater biofilms could play an important role in the migration and transformation of organic pollutants, especially under illumination conditions [8, 9].
- Materials and Methods: The two biofilms come from different regions. Are there congenital differences between the two biofilms due to their geographical location? In the view of the reviewer, the congenital difference may attenuate the effect of water quality?
We appreciate the reviewer’s valuable comment. It might be true that congenital differences exist between the two biofilms due to their geographical location. Nonetheless, in the present study, the freshwater biofilms were rinsed and washed to remove the residues prior to batch experiments, which was followed by a series of published works evaluating the effects of biofilms from distinct origins on pollutant bioaccumulation or degradation.
- Results and Discussion: Please appropriately add data description and analysis of Table 1 and Table 2 in section 3.3.
Accepting the reviewer’s valuable comment, the related expressions have been revised as shown below:
Table S4 provides the compositional properties of EPS derived from both fresh-water biofilms. For an EPS solution at a TOC concentration of 50 mgC/L, the EPS de-rived from river biofilm contained 94.3±16.4 mg/L carbohydrates, 13.2±0.2 mg/L pro-teins, and 26.7±0.1 mg/L humic substances, respectively. While for pond biofilm EPS at the same TOC content, the concentrations of carbohydrates, proteins, and humic sub-stances were 72.6±27.5 mg/L, 4.8±1.0 mg/L and 15.5±0.7 mg/L, respectively. The content of proteins and humic substances in river biofilm EPS were 2.75 and 1.73 times higher than in pond biofilm EPS.
An integrated spectral approach was employed to decipher the differences be-tween EPS fractions derived from both freshwater biofilms. Table S5 provides spectral parameters of both EPS fractions. River biofilm EPS had higher a(355), SUVA254 and SUVA260 values than pond biofilm EPS, implying that the EPS from the biofilm collect-ed in effluent-dominated rivers contained more fractions with high aromaticity, and the fractions were more hydrophobic [35, 36]. The value of E2/E3 was reported to be significantly negatively correlated to the apparent molecular weight and aromatic condensation degrees. The higher E2/E3 value of pond biofilm EPS suggested that the EPS molecules derived from natural pond biofilm contained more high-molecular-weight fractions and aromatic condensation degrees. River biofilm EPS had a higher HIX value compared to those of the pond biofilm EPS. The results sug-gested that river biofilm produce more humified materials [37]. The BIX value of river biofilm EPS was with the range of 0.8 to 1, demonstrating that the biopolymers were predominantly originated from an autochthonous source. While the BIX of pond bio-film EPS was higher than 1, suggesting that structural plant, bacterial and algal resi-dues comprised the major sources.
- Please double-check the format and further improve the language.
Modifications have been revised as requested.

Reviewer 2 Report
The authors studied the photodegradation of methyl orange and bisphenol A by different freshwater biofilms. The work is interesting, however, some questions should be addressed before it can be accepted.
1. Full name of abbreviations should be provided. For example, 3EPS*, 3EOM*, SBR.
2. Full spectra of 13 C-NMR that mentioned in Table 2 should be provided and discussed carefully.
Author Response
The authors studied the photodegradation of methyl orange and bisphenol A by different freshwater biofilms. The work is interesting, however, some questions should be addressed before it can be accepted.
- Full name of abbreviations should be provided. For example, 3EPS*, 3EOM*, SBR.
We appreciate the reviewer’s valuable comment and the full names of the abbreviations have been added as marked in red. 3EPS*, 3EOM* and SBR refer to the triplet excited-states EPS species, the triplet excited-states species and the sequencing batch reactor respectively.
- Full spectra of 13 C-NMR that mentioned in Table 2 should be provided and discussed carefully.
Accepting the reviewer’s valuable comment, the full spectra of 13 C-NMR that mentioned in Table 2 were provided and discussed
“The freeze-dried EPS samples were analyzed by 13C NMR to characterize the chemical functional groups. Table 2 shows the contributions of each functional group in both EPS fractions via NMR analysis, including the alkyl, methoxyl, crbohydrate, ar-yl, O-aryl, carboxyl, carbonyl, aromatic carbon, aliphatic carbon and polar carbon frac-tions. Table 2 demonstrates a relatively large contribution of alkyl (0–45 ppm) and car-bohydrate (63–93 ppm) fractions, ranging from 34.6% to 41.3% and from 37.2% to 38.1%, respectively. Compared with pond biofilm EPS, EPS derived from river biofilm con-tained higher amounts aromatic carbon (7.9% compared to 2.1%) and polar carbon (61.6% compared to 56.5%), suggesting that river biofilm EPS were more aromatic and hydro-philic, consistent with the results of spectral analysis in Figure 5 and Table S5.”
Reviewer 3 Report
The article is interesting and achieved results are good and meet aims. The article proved that two target Freshwater biofilms could accelerate the photodegradation of both 448 pollutants, the direct photodegradation rate of MO and BPA was 8.7 times and 5.6 time 449 increased, respectively when dosed with river biofilms.
Only one concerns is about the statistical calculations that should be provided.
Author Response
The article is interesting and achieved results are good and meet aims. The article proved that two target Freshwater biofilms could accelerate the photodegradation of both pollutants, the direct photodegradation rate of MO and BPA was 8.7 times and 5.6 time increased, respectively when dosed with river biofilms.
Only one concerns is about the statistical calculations that should be provided.
We appreciate the reviewer’s valuable comment, The protocols of the statistical calculations of the results are provided in Section 2.6:
2.6. Analytical methods
For each photochemical reaction experiment, aliquots of 2.0 mL filtrated samples were collected at different time intervals, i.e., 0, 5, 10, 15, 30, 45 and 60 min, from the solution. The content of MO was measured via UV spectra at a wavenumber of 465 nm. The contents of BPA were determined by high performance liquid chromatography (1260 Infinity, Agilent Co., USA) equipped with a 5 μm×4 mm×150 mm Eclipse XDB-C18 column set at 35 oC and a UV detector at a wavenumber of 230 nm. The mobile phase was a mixture of 10 mM H2O/acetonitrile 50:50 (v/v) at a flow rate of 1 mL/min. The photodegradation rate constants of MO and BPA followed pseudo-first-order depletion were obtained by linear regression following eq1:
where Kobs, t, C and C0 refer to the pseudo-first-order depletion constant (min-1), reaction time (min), pollutant concentration during reaction (mg/L) and initial concentration of pollutant (mg/L), respectively..
